# Multi-Agent Deep Reinforcement Learning for Online 3D Human Poses Estimation

**Zhen Fan** [1,2], **Xiu Li** [1,2] and **Yipeng Li** [2,*]

1   Graduate School at Shenzhen, Tsinghua University, Shenzhen 518057, China;
    fanz14@mails.tsinghua.edu.cn (Z.F.); li.xiu@sz.tsinghua.edu.cn (X.L.)
2   Department of Automation, Tsinghua University, Beijing 100084, China
*   Correspondence: liep@tsinghua.edu.cn

**Abstract:** Most multi-view based human pose estimation techniques assume the cameras are fixed. While in dynamic scenes, the cameras should be able to move and seek the best views to avoid occlusions and extract 3D information of the target collaboratively. In this paper, we address the problem of online view selection for a fixed number of cameras to estimate multi-person 3D poses actively. The proposed method exploits a distributed multi-agent based deep reinforcement learning framework, where each camera is modeled as an agent, to optimize the action of all the cameras. An inter-agent communication protocol was developed to transfer the cameras' relative positions between agents for better collaboration. Experiments on the Panoptic dataset show that our method outperforms other view selection methods by a large margin given an identical number of cameras. To the best of our knowledge, our method is the first to address online active multi-view 3D pose estimation with multi-agent reinforcement learning.

**Keywords:** multi-person; online poses estimation; multi-agent reinforcement learning; consensus; panoptic dome





## 1. Introduction

Human pose estimation has attracted much attention in recent years due to its wide human-centric applications [1–3]. 3D pose estimation methods provide 3D positions of human joints by using a single-view image [4] or multi-view images [5,6]. Single-view based 3D pose reconstruction is an ill-posed problem, so current literature estimates 3D joints' positions by the prior knowledge learned from data. Multi-view based approaches estimate the poses by triangulation. Most of these methods treat the camera as fixed, suffering from invisibility caused by occlusion and inaccuracy caused by extreme viewpoints. Many works try to select sparse views to reconstruct human poses from a dense variety of viewpoints like the Panoptic massive camera grid [7,8]. They use a deep reinforcement learning-based method to select informative viewpoints for higher estimation accuracy. This method leverages a single agent selecting views of the same frame in multiple steps, during which it mandates fixed wall time or static human. This is impractical in a live environment. Specifically, all agents must observe the environment simultaneously at each time step, take actions and step to the next frame together. Therefore, we need to optimize their actions in an online manner rather than select views in a 'time-freeze' mode. A typical scenario is that we have a fixed number of moving cameras, i.e., live broadcasting sports with multiple cameras, and we need to learn a policy to control the movement of each camera on the current observations of the dynamic scene. Given the requirement for efficiency of an online scenario in the real world, the moving cost of the camera often matters due to physical limitations.

In this paper, we propose a distributed **M**ulti-**A**gent **O**nline Pose estimation architecture (*MAO-Pose*) to control a fixed number of moving cameras to estimate 3D human poses actively. In this distributed system, each camera has its individual observation and learns

from the total reward. The total reward consists of the view quality and penalty term. We use the joints visibility as the view quality measure instead of the pose reconstruction error, and thus the network can be trained in a self-supervised way without the requirement of ground truth pose annotation. The penalty term punishes agents when they select identical views to encourage the diversity of view selection. Camera movement constraints can be included in this term as well to increase the continuity between consecutive frames.

We also introduced an inter-agent communication mechanism, *Consensus*, to ensure the collaboration of multiple agents. We chose only to share the position information between agents, which is effective and of low computational cost. The agents plan their actions sequentially in a pre-defined order. The former agent will inform the following agents of its movement and the latter agents will optimize their action plans with awareness of the expected new positions of the former agents. After this coordination process, all agents will take actions simultaneously to enter the next time step.

We evaluate our method on the Panoptic dataset [9], which has a dense variety of views forming a spherical dome. These dense views serve as potential viewpoints for our system. The results show that our method assigns valuable views for the cameras and can move the cameras with lower movement cost, if needed. Various ablation studies are provided to corroborate the efficacy of the communication protocol, *Consensus*, and offer comparisons of different reward designs, i.e., whether to use the supervised signal as a view quality measure and enforce movement constraints in the penalty term.

To summarize, our contributions are threefold:

- To the best of our knowledge, we are the first to address the task of online multi-person 3D pose estimation with multiple active views.
- We propose a distributed multi-agent pose estimation architecture (*MAO-Pose*) to control a fixed number of moving cameras to estimate 3D human poses actively.
- We explore different reward design choices: self-supervised vs. supervised view quality measure, and camera movement constraints when camera continuity is required.

## 2. Related Work

**Pose Estimation from Fixed Views** Most vision-based methods estimate 3D human pose with multiple fixed and well-calibrated camera systems. After 2D pose detection from images, various algorithms [9,10] are exploited to find cross-view joints correspondence and triangulate all 2D joints to 3D in an epipolar geometry framework. Taking massive view images as input, Joo et al. [9] vote for the 3D position of every joint and the most likely connection between them at each time instance. Considering the environment occlusion in multi-person scenarios, Zhang et al. [11] propose a 4D association graph which introduced temporal tracking of all joints to ameliorate the missing joint problem. However, such a statically mounted capture system is inapplicable in outdoor scenarios. Recent studies have focused on lifting 2D poses to 3D directly from RGB image(s) using different CNN (convolutional neural networks) based network structures [3,12–16]. Given reliable 2D pose detection results developed in [1,17–19], learning methods could achieve accurate results in real-time even under partial occlusion of the target human [20]. However, such learning methods heavily rely on the observation quality which could not be guaranteed by casually captured images, especially in long-term tasks.

**Active Pose Estimation** To overcome the spatial and temporal limitations simultaneously, an increasing number of active camera based motion capture methods have been developed in recent literatures [21–23]. Taking a camera-equipped and manually controlled drone as a recording device, Zhou et al. [24] formulated this task as a nonrigid structure from motion (NRSFM) problem and recovered human motion from the video recorded in fast varying viewpoints. Naegeli [25] propose an environment-independent approach, *Flycon*, to reconstruct the target human pose with two camera-equipped drones. This approach optimizes the drone states and target human pose jointly and could track human motion in indoor and outdoor environments over a long time and distance. Considering the cinematography requirements for action scenes, Kiciroglu [21] predicts the best view of

a single camera in the future frame to maximize the accuracy of 3D human pose estimation while considering the motion limitations. However, this work is limited to pose estimation of a single person with a single camera. Huang [26] designed an autonomous cinematography system to obtain the most informative viewpoint that respects the physical moving constraints of a flying robot equipped with one binocular camera. However, existing flying motion capture methods are not applicable to multiple human scenes without considering occlusions between human bodies.

Recently, much research has concentrated on deep reinforcement learning methods for policy-based tasks, such as view planning for 3D object model recovery [27,28] and action prediction [29–31]. Multi-agent framework is also used in some relevant domains including multi-robot motion planning [32,33] and environment mapping [34]. An RL-based multi-person 3D pose estimation method, *ACTOR*, is committed in [7] to solve the active view selection problem. In contrast to *ACTOR*, *Pose-DRL* [8] introduces 3D fusion accuracy as the viewpoint selection reward and demonstrates the effectiveness in single and multi-person settings. These methods select multiple views for a frame in multiple steps, during which the wall time must be constant. Thus, they must rely on the assumption that people inside the scene remain static, which is impractical in real environments. Compared with previous methods, our work could select multiple views simultaneously, allowing for real-time application, and it achieves better performance in dynamic environments using multi-agent framework.

### 3. *MAO-Pose*: Multi-Agent for Online Pose Estimation

In this section, we demonstrate our distributed multi-agent framework for active 3D human pose estimation. We first formulate the problem of active 3D human pose estimation (illustrated in Figure 1) in Section 3.1, together with how the Markov Decision Process (MDP) proceeds in our setup; then, introduce details of *MAO-Pose*. The representation of state and action is presented in Section 3.2. The design of annotation-free reward signal is introduced in Section 3.4, which can train the agents efficiently to triangulate joints of all people in the scene while considering possible view duplication at the same time. Camera movement constraints can be easily integrated to our framework if inter-frame continuity is required. The inter-agent communication mechanism is presented in Section 3.5, which encourages agents to work collaboratively. We also provide an explicit optional prediction module, Prophet, in Section 3.6, which predicts the future state of the observation for policy learning in a dynamic scene. An overview of *MAO-Pose* is shown in Figure 2.

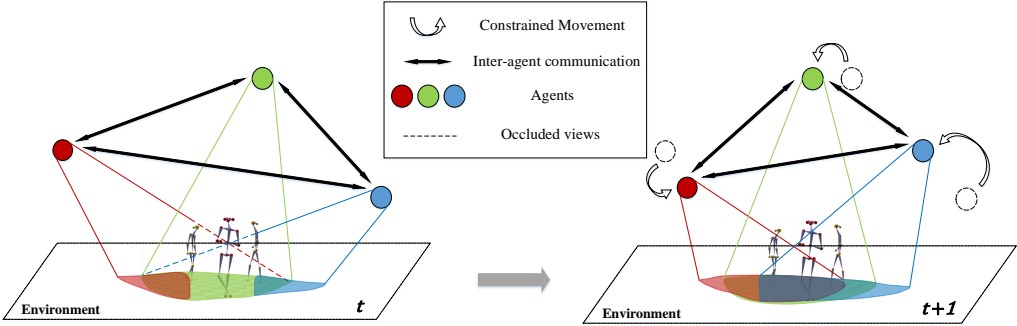

**Figure 1.** Illustration of how agents operate online view selection from time step $t$ to $t + 1$, here shown for a triple-agent scenario. At time step $t$, agents observe the environment with multiple views among which some are occluded by human bodies (shown as colored dash lines). To avoid the current occlusion and get a more accurate estimation of human poses, agents jointly move to new viewpoints at the next time step $t + 1$, which are generated by the learnt policy.

### 3.1. Online Active Pose Estimation

The online active 3D human pose estimation problem is defined in the context of $n$ agents moving in a 3D space while observing the scene from different views. The

ultimate goal of agents is getting to the best viewpoints at each time step given the current observations, and extracting 3D poses of $m$ people in the scene collaboratively. A typical scenario is illustrated in Figure 1.

In particular, at each time step $t$, the $i$-th agent has an observation $V_i^t$ (typically an image containing multiple people) of the scene from its current position. Then, the agent samples an action $A_i^t$ from a stochastic policy $\pi_{\theta_i}$ parameterized by $\theta_i$. Once all agents have taken actions, the system collects 2D human poses $\mathbf{X}^t$ estimated by each agent and then steps to the next frame at time $t + 1$. The process repeats until the end of the sequence of frames.

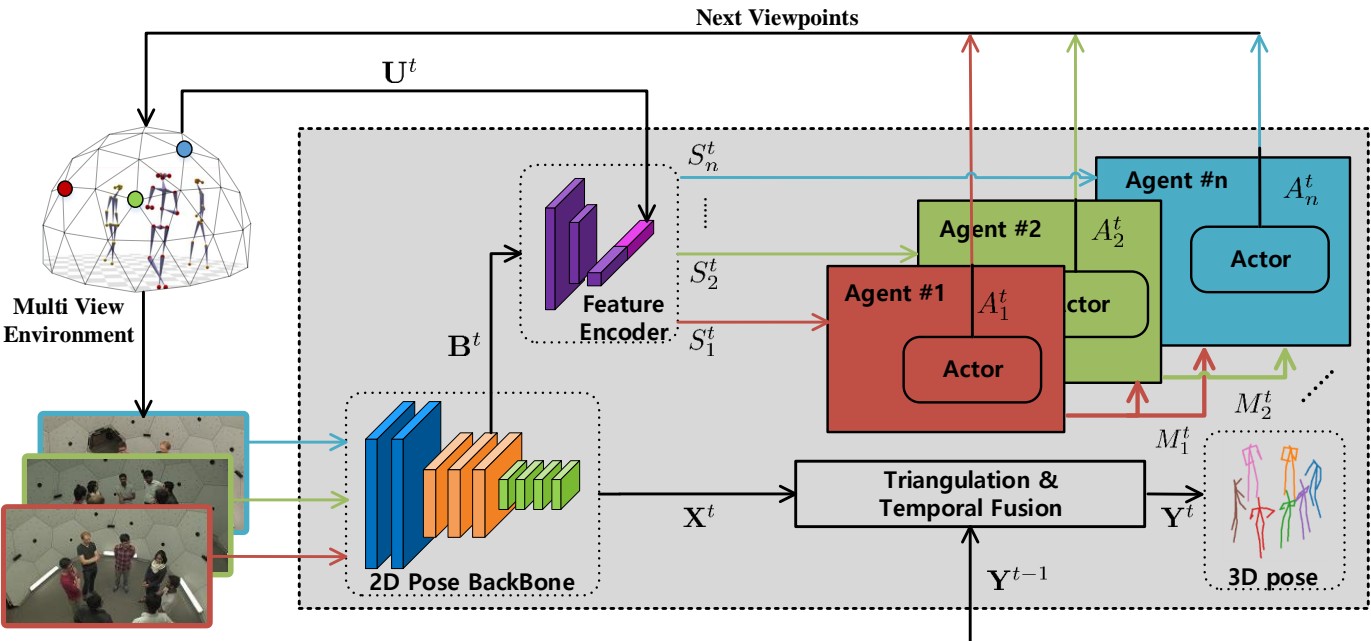

**Figure 2.** Overview of *MAO-Pose*. Each agent, labeled with different a color, corresponds to each camera with individual observation. The cameras are placed in the multi-view spherical dome environment. At time step $t$, the system observes the environment subject to the current positions of agents, receiving multiple images that are then passed through a 2D pose estimator. A triangulation and temporal fusion module takes the 2D pose estimates $\mathbf{X}^t$ yielded by 2D pose backbone, combining previous estimates $\mathbf{Y}^{t-1}$ to generate final 3D estimates $\mathbf{Y}^t$. Meanwhile, a feature encoder takes feature blobs $\mathbf{B}^t$ from the intermediate layer of 2D pose backbone and auxiliary information $\mathbf{U}^t$ into state vector $\mathbf{S}^t$. The $i$-th agent produce the camera moving action $A_i^t$ from $S_i^t$ based on policy network *Actor*. Agents can utilize the *Consensus* protocol to send its planned action message $M_i^t$ to other agents for collaboration. Once all the agents have output their actions, they interact with the environment concomitantly, advancing to next time step $t + 1$.

We assume that the correspondence of 2D poses between frames is accurate, which means that there is no need to perform re-identification of human bodies over frames. Thus, at time step $t$, the estimated 3D poses $\mathbf{Y}^t$ are directly triangulated from 2D poses $\mathbf{X}^t$, and temporally fused with the estimates $\mathbf{Y}^{t-1}$ from the previous time step to refine estimated poses.

Note that, we idealize our agent–environment setup using Panoptic [9] as it captures time-synchronized HD videos in a spherical dome with multiple people and views. The candidate viewpoints are sampled on a spherical dome which is split into $w \times h$ bins along azimuth and elevation directions, respectively.

An OpenPose [35] based backbone is employed to extract feature instances $\mathbf{B}^t = [B_1^t, B_2^t, \ldots, B_n^t]$ from observations $\mathbf{V}^t = [V_1^t, V_2^t, \ldots, V_n^t]$, yielding 2D pose estimates $\mathbf{X}^t = [X_1^t, X_2^t, \ldots, X_m^t]$ for all visible people. The 3D poses estimation $\mathbf{Y}^t$ generated by triangulation over 2D poses $\mathbf{X}^t$ is referred to as $\mathbf{Y}^t = [y_1^t, y_2^t, \ldots, y_m^t]$ at each time step $t$, where $m$ is the number of people estimated.

### 3.2. State-Action Representation

In this section, while describing the state and action representations, we must emphasize that the distributed agents act over the time sequence which can be formulated as a multi-agent decision process. The state of the $i$-th agent is represented as a tuple $S_i^t = (B_i^t, U_i^t)$, where $B_i^t$ is the feature map from an Openpose [35] based backbone and $U_i^t$ is auxiliary information consists of $C_{1,i}^t$, $C_2^t$, $N_1^t$, $N_2^t$. More specifically, $C_{1,i}^t \in \mathbb{N}^{w \times h}$ is a camera state counter for the $i$-th agent, $C_{1,i}^t(j,k)$ increases once the bin at location $j,k$ is selected by the $i$-th agent. $C_2^t \in \mathbb{N}^{w \times h}$ is a fixed matrix denoting available camera locations on the spherical dome. $N_1^t \in \mathbb{N}$ equates to number of people detected at current time step $t$ while $N_2^t$ indicates that whether the current frame is the initial one of a sequence.

We employed a deep stochastic policy $\pi_{\theta_i}(A_i^t | S_i^t)$ to guide the agents taking action, where $\theta_i$ denotes the policy parameters for the $i$-th agent. The action is defined as $A_i^t = (\phi_a^t, \phi_e^t)$, where $(\phi_a^t, \phi_e^t)$ is the azimuth–elevation angle pair indicating the relative camera movement along azimuth and elevation, respectively. We sampled these angles from the periodical Von Mises distribution. The structure of the policy network is shared by all agents while each agent has its own parameters, i.e., $\pi_\theta = [\pi_{\theta_1}, \pi_{\theta_2}, \dots, \pi_{\theta_n}]$.

### 3.3. MAO-Pose Agent

Figure 3 illustrates the structure of the policy network of the $i$-th agent. Taking the feature map $B_i^t$ as input, the shared convolution layer consists of Conv($128 - 3 \times 3 - 8$)-Conv($8 - 3 \times 3 - 4$) where Conv($I - K \times K - O$) represents convolution layer has $I$ input channels and $O$ output channels with kernel size of $K \times K$. In order to down-sample sufficiently, we employed max pooling with kernel size of $2 \times 2$ after the first convolution layer. We then introduced fully-connected layers after the flattened output of the feature encoder concatenated with auxiliary information $U_i^t$.

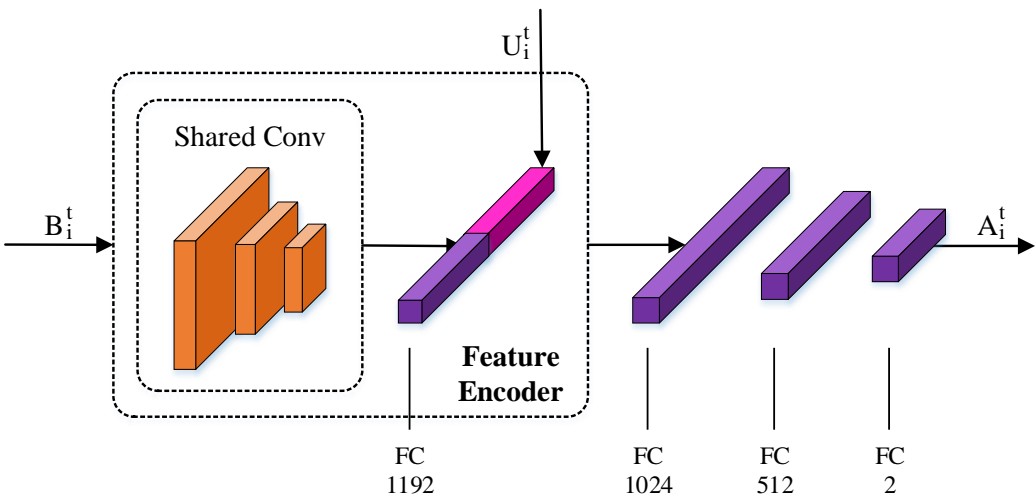

**Figure 3.** Structure of the policy network of the $i$-th agent.

### 3.4. Reward Design

Our objective is to select more valuable views and avoid unwanted occasions, in our case, multiple agents selecting the identical views.

In order to maintain a distributed architecture, each agent receives its reward individually. The reward of agent $i$ at time step $t$ is a sum of two terms,

$$r_i^t = Q^t + P_i^t \tag{1}$$

where $Q^t$ is the quality of the selected views. We set $Q^t = \min_k \mathcal{J}_k^t$, where $\mathcal{J}_k^t$ is the proportion of joints visible for the $k$-th person at time step $t$, which represents the visibility

of 3D joints in all views. A joint is visible if it could be seen from at least two views. To encourage agents working collaboratively, $Q^t$ is shared across agents at time step $t$.

$P_i^t$ is the penalty term and $P_i = w_{col}\tau_{col}^t$, where the weighting factor $w_{col} = 0.2$ and $\tau_{col}^t$ is set to 1 when two or more agents selects the same view. The penalty term $P_i^t$ is designed to simulate collisions between moving cameras on the idealized setup, Panoptic Studio, where cameras are distributed on a spherical dome. Intuitively, this term not only simulates potential collisions, but also plays an important role to improve the qualities of viewpoints and encourage exploration at the training stage. According to our experiments, agents do not separate enough without this penalty term, leading to a poor performance on reconstruction error. From the perspective of view-selection, such a penalty forces the agents to decline repeated observations which are not informative and bring no benefit to triangulation accuracy. On the other hand, in terms of reinforcement learning, such a penalty encourages each agent to explore novel viewpoints during the early training phase, helping the system getting away from local optima of MPJPE. Furthermore, note that, *MAO-Pose* aims to generate an online policy for 3D human pose estimation instead of a full stack controller. When extending MAO-Pose to real-world scenarios, a possible solution for collision avoidance is to set an exclusive sphere for each agent, whose radius could be the minimum distance between the two viewpoints on the Panoptic dataset.

Policy gradients are used to train policy parameters $\theta_i$ of each agent in which we maximize expected cumulative rewards with the objective

$$J(\theta_i) = \mathbb{E}_{s \sim \pi_{\theta_i}} [\sum_{t=1}^{|s|} r^t] \tag{2}$$

where $s$ denotes state-action trajectories generated on policy $\pi_{\theta_i}$. We used REINFORCE [36] to approximate the gradient of this objective.

### 3.5. Inter-Agent Communication

We introduced an efficient inter-agent communication mechanism, *Consensus*, to encourage the agents to work together. In our distributed multi-agent framework, the agents do not share any observations. *Consensus* works in the action procedure. Each agent is assigned an ID beforehand and they will plan action in the order of their IDs. When agent #$i$ finishes planning, it will inform agents with ID bigger than $i$ of its planned action. Furthermore, these notified agents will record the plan of agent #$i$ by incrementing the camera counter $C_{1,i}^t$ in their local coordinate. The detailed mechanism of *Consensus* is listed in Algorithm 1. With the communication protocol, agents can coordinate their actions before interacting with the environment, resulting in better choices with the awareness of further system states.

---

**Algorithm 1** Consensus Protocol

---

1: **procedure** CONSENSUS(*agents*) ▷ Each agent is given an ID #1, #2, $\cdots$ , #$n$ beforehand
2:　　CLEAR(*agent*[1].*state*.$C_{1,1}^t$)
3:　　*action_list* $\leftarrow \varnothing$
4:　　**for** $i \in \{1, 2, \cdots, n\}$ **do**
5:　　　　*angle* $\leftarrow$ PLANACTION(*agent*[$i$])　　　　　▷ Output an action via policy network
6:　　　　*action_list* $\leftarrow$ *action_list* $\cup \{angle\}$
7:　　　　**for** $j \in \{i+1, \cdots, n\}$ **do**　　　　　▷ Agent #$i$ sends message to later ones
8:　　　　　　*local_angle* $\leftarrow$ CONVERTTOLOCAL(*angle*) ▷ Angles relative to current agent
9:　　　　　　INCREMENTCOUNTER(*agent*[$j$].*state*.$C_{1,j}^t$, *local_angle*)
10:　　COMMITACTION(*action_list*) ▷ All agents reach a consensus and act simultaneously

---

### 3.6. Prediction Module Prophet

In this task, we need to assign the cameras to the optimal positions to capture the people in the next time step based on the observation of the current time step. However,

in a dynamic environment, policy based on the current state is inaccurate. Hence, we introduce an optional explicit prediction module, *Prophet*, to predict the state in the next time step. This module is connected before Actor and shares the same structure among all the agents. Here we consider the state of *i*-th agent.

Modified from [37], *Prophet* consists of three modules parameterized by $\theta = (\theta^{value}, \theta^{out}, \theta^{trans})$: (i) **Value** module $f_\theta^{value}$ estimates the value of state $S_i^t$. A simple fully connected network consists of three layers is used as value module. (ii) **Outcome** module $f_\theta^{out}$ predicts reward $r_{p,i}^t$ of state $S_i^t$. (iii) **Transition** module $f_\theta^{trans}$ transforms from state $S_i^t$ to predicted state $S_{p,i}^{t+1}$ at the next time step. We employed a simple fully-connected network and add a residual connection from the previous state to the next state so that the transition module learns the difference from previous states to future states.

Here we demonstrate how *Prophet* performs 1-step prediction by composing the above modules: $f_\theta^{prophet} : S_i^t \longrightarrow S_{p,i}^{t+1}, r_{p,i}^t, V(S_{p,i}^{t+1})$. Note that, though only the **Transition** module is needed (to predict $S_{p,i}^{t+1}$) for evaluation, **Value** and **Outcome** are required during the training process of *Prophet*.

As shown in Figure 4, the **Transition** module of *Prophet* consists of two fully-connected layers whose dimension is equal to the dimension of state $S^t$. The **Value** module consists of FC(1192)-FC(512)-FC(1) where FC(N) represents a fully-connected layer with N hidden units. Furthermore, the **Outcome** module shares the same structure with the **Value** module but has independent parameters. An exponential linear unit (ELU) [38] was used as an activation function for all architectures.

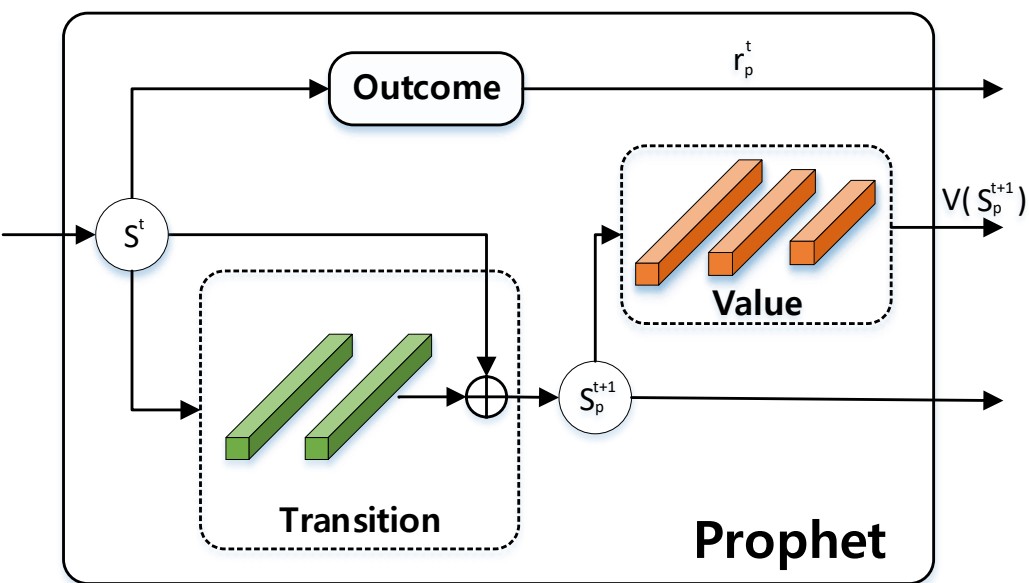

**Figure 4.** Structure of prediction module, *Prophet*.

## 4. Results

### 4.1. Implementation Details

*MAO-Pose* is implemented on top of an optimized version of Lightweight Open-Pose [35]. Theoretically, our framework can be easily adapted to any 2D pose estimator with feature backbones. The 3D pose estimation is achieved by triangulating each pair of estimated 2D poses via multi-view geometry and taking the median of all 3D candidates. For temporal fusion, we used One Euro Filter [39], a speed-based low-pass filter to refine estimated poses. The missing joints on the current frame are inferred from previous frames, if available. In case there is no valid previous estimation, we set the reconstruction error a high value (500 mm) to avoid NaN values.

### 4.1.1. Dataset

We consider multi-people scenes (*Mafia*, *Ultimatum* and *Haggling* in the Panoptic dataset), where occlusions occur frequently, posing a challenge for the agents to reconstruct the 3D pose. The Panoptic dataset consists of 30 FPS time-synchronized multi-view videos. To ensure the convenient extraction of single frames and agree with maximum control frequency 10 Hz, we convert them to JPEG images and temporally downsample frame rate to 10 FPS beforehand. We only use HD cameras, the number of which is about 31 per scene, since they provide better image quality and their locations are not too sparse to hinder the decision process of agents. We randomly chose 10 scenes out of *Mafia* for training, 6 and 21 scenes from all scenes for validation and test, respectively. The splits have no overlap between each other. We trained on limited scenes but tested on all three kinds to force agents to learn a fairly generalized policy. Table 1 shows the size of the train, validation and test splits. The *Haggling* scene is completely invisible during training (with validation sets for early stopping).

**Table 1.** The number of frames categorized by scene name and split type. Note *Mafia* and *Ultimatum* are denser scenes with 7 people, while *Haggling* comes with 3 people.

| Scene | Train | Validation | Test |
|:---:|:---:|:---:|:---:|
| **Mafia** | 1741 | 930 | 1088 |
| **Ultimatum** | N/A | 140 | 1925 |
| **Haggling** | N/A | N/A | 1609 |
| **Total** | 1741 | 1070 | 4622 |

### 4.1.2. Evaluation Metrics

We used two metrics to evaluate the performance. *Mean per joint position error* (MPJPE) indicates the pose reconstruction error w.r.t. the ground truth. *Camera distance* (CamDist), defined as the geodesic distance between camera positions in consecutive frames on the dome-like spherical surface of the Panoptic dataset, denotes the moving cost of cameras.

### 4.1.3. Baselines

We implemented four baselines to compare with our agents. All methods utilized completely identical 2D pose estimator, temporal fusion, matching, triangulation method and multi-agent environments for fairness. The baselines are as follows: (i) *Random*: randomly selects $N$ cameras in a given frame; (ii) *AngleSpread*: spreads cameras by choosing cameras that have $2\pi/N$ of azimuth difference. At each azimuth angle, a random elevation angle is sampled; (iii) *Greedy*: greedily choose unvisited cameras that minimize pose reconstruction error most. This exhaustive search method is impractically slow compared with other baselines, yet it exhibits the nearly optimal lower bounds of reconstruction error; (iv) *ACTOR*: a deep-RL architecture we implemented according to [7]. Note that as a single-agent paradigm, *ACTOR* requires multiple time steps to select multiple views, which must *pause* the wall time before it selects enough cameras.

For all the baselines, since they do not model different views as individual agents, we used the Hungarian algorithm to match the cameras selected in the current and previous frame to greedily minimize the camera distance between frames and simulate the behavior of multiple agents. However, this matching mechanism is disabled for *MAO-Pose* agents, as they are not fungible in our context.

### 4.1.4. Training

We used sampled returns to approximate value function, with discount factor $\gamma = 0.9$ for future rewards. Each agent receives the rewards individually and normalizes the discounted rewards during each training step to reduce variance and increase stability. The objectives are optimized via the Adam optimizer with a fixed learning rate $10^{-6}$. We trained the agents together for 80 k episodes, annealing the concentration parameters

($\kappa_a$, $\kappa_e$) of Von Mises distribution linearly with rate ($3 \times 10^{-4}$, $5 \times 10^{-4}$), which ensures adequate exploration at first and then becomes more deterministic as the agents grow more sophisticated. We utilized validation sets during training to perform early stops when necessary. The average reward each agent receives during the training process is shown in Figure 5.

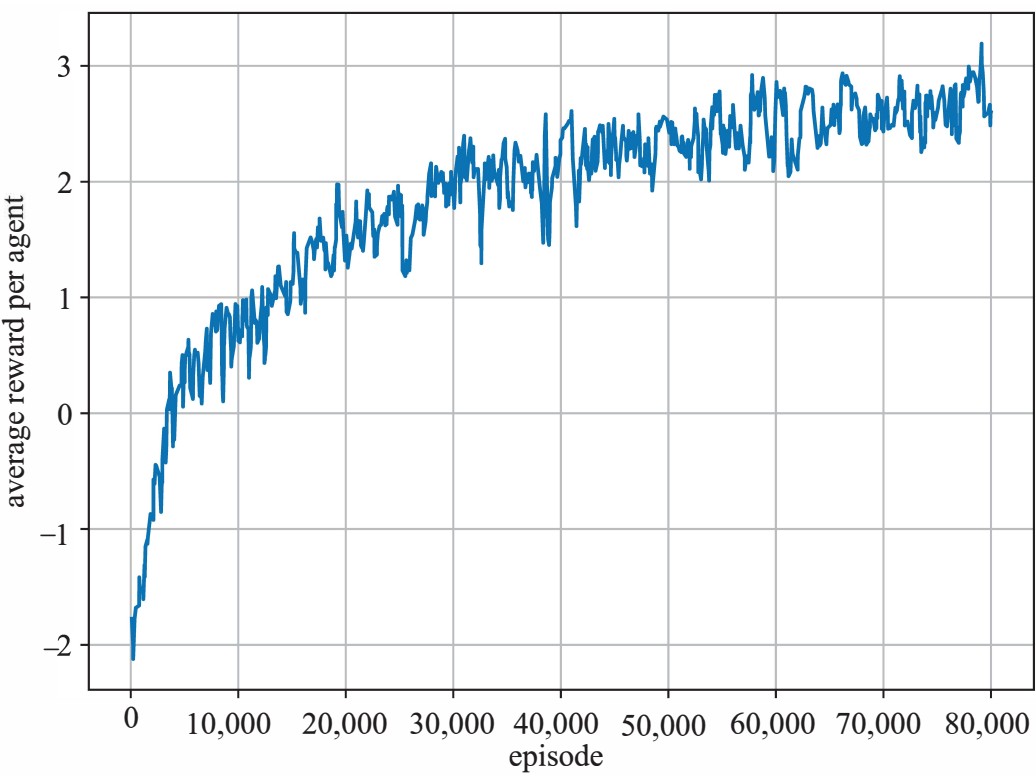

**Figure 5.** Average rewards per agent with *Consensus*.

### 4.2. Main Results

Our *MAO-Pose* is trained with 5 agents in total, *Consensus* enabled and FPS set to 10 and is compared with four baselines on the Panoptic test dataset. In Tables 3 and 4, we report the numerical values of MPJPE and CamDist for *MAO-Pose*. The results are averaged across the union of three scenes (*Mafia*, *Ultimatum* and *Haggling*) and 5 random seeds, with standard deviation in the bracket. In terms of MPJPE, our model significantly outperforms other methods and is very close to *Greedy*, which is nearly a theoretically optimal solution. In terms of CamDist, our model with movement constraints (*MAO-Pose*-Cons) achieves the lowest moving cost.

Since *MAO-Pose* operates on frame sequences, we want to evaluate its performance temporally. Figure 6 illustrates MPJPE and camera distance on *Mafia* (7 people) and *Haggling* (3 people) scenes, to compare the performance in environments with different human densities. *MAO-Pose* exhibits more stability during the entire sequence without large peak values while keeping both metrics low enough to outperform all other baselines. Here, we visualized the estimated 3D poses and their reprojected 2D poses in each view the agents have selected. Please refer to Figure 7 for detail.

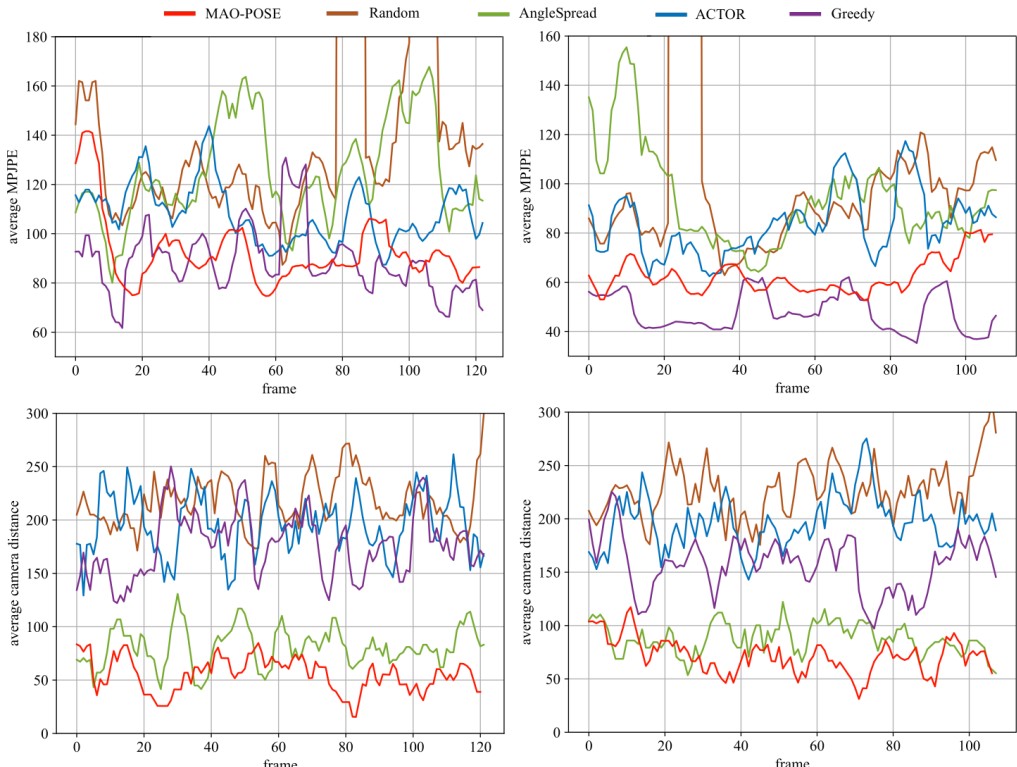

**Figure 6.** Sequential performance in the scene from *Mafia* (**left** column) and *Haggling* (**right** column), respectively. The metric: top row is MPJPE and bottom row is camera distance. *MAO-Pose* is able to achieve both a lower camera moving distance and reconstruction error, and works more stably during the whole sequence. Though *Greedy* can achieve lower average MPJPE, its stability is not sufficient, thus overall *MAO-Pose* outperforms all other baselines.

### 4.3. Ablation Studies

**The number of agents:** As the results showed in Table 3, with an increasing number of views, *MAO-Pose* achieves lower estimation error, since more valid views are used for reconstruction. However, in Table 4 CamDist increases slightly, especially for *MAO-Pose-Cons*. This is because, with more cameras in the system, the collision probability is higher. The cameras may move a larger distance to avoid collision.

*Consensus:* Without the *Consensus* module, all the agents take action independently without knowing the action of other agents. The result is shown in Tables 3 and 4 as *MAO-Pose-noComm*. Both MPJPE and CamDist metrics decrease significantly without *Consensus* module, which proves that our *Consensus* module enables the agents to work together collaboratively with more information shared in between.

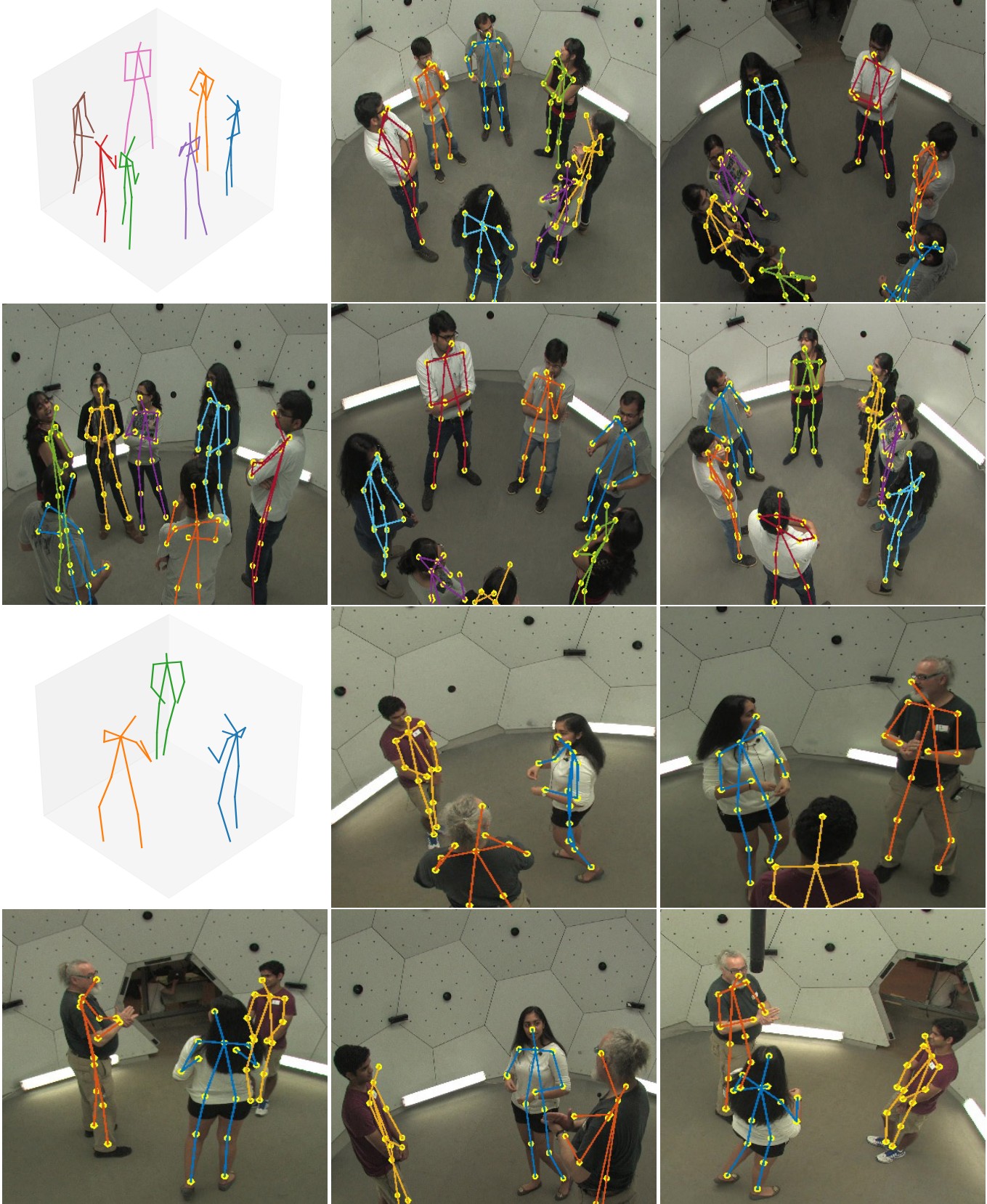

**Figure 7.** Experiment results on two different types of multi-people scenes. The 2D poses in each view are obtained by reprojecting 3D poses to camera coordinates via camera parameters. **Top**: Dense scene with 7 people in total. *MAO-Pose* is able to reconstruct all the peoples using 5 views. **Bottom**: Sparser scene with only 3 people.

**Movement constraints:** Since our agents operate on temporal sequences, it would be beneficial to take inter-frame continuity into account. It is desirable if camera movements between two consecutive frames are not too large. To encourage the agent to move with lower cost, a piece-wise linear function $w_{dis}\tau_{dis}$ can be added to the penalty term $P_i$ in Equation (1), which is based on moving constraints to punish large camera movements between two frames:

$$
\tau_{dis}(x) = \begin{cases} 0, & x \leq \lambda_1 \\ \dfrac{x - \lambda_1}{\lambda_2 - \lambda_1}, & \lambda_1 < x \leq \lambda_2 \\ \mu \dfrac{x - \lambda_2}{\pi r - \lambda_2} + 1, & x > \lambda_2 \end{cases}
\tag{3}
$$

where weighting factor $w_{dis} = -0.2$, and $x$ is the geodesic distance of camera position between time step $t$ and $t + 1$ on the spherical dome in the Panoptic dataset. $\lambda_1$, $\lambda_2$ are predefined parameters of sensitivity and $r$ is the radius of the dome. $\mu$ is a scaling factor typically larger than 1. We set $\lambda_1 = 5.0$ (smaller than the minimum distance of cameras in the dataset) to encourage local exploration, within which agents will not be penalized. $\lambda_2 = 100.0$ and $\mu = 10$ to enforce high penalty for large camera distances. $\lambda_2$ is set according to the experiments in Table 2. It is worth noting that camera distance and reconstruction error are partially adversarial. Constraining camera distance can result in a limited search area and thus deter the accuracy. We chose $\lambda_2 = 100$ since its error is close to that of $\lambda_2 = 700$, with a much lower camera distance, which is an acceptable trade-off between the two metrics.

**Table 2.** The results on different sets of $\lambda_2$. The final $\lambda_2$ is in bold.

| $\lambda_2$ | 50 | **100** | 150 | 200 | 300 | 500 | 700 |
|---|---|---|---|---|---|---|---|
| MPJPE | 97.93 | 95.88 | 96.89 | 95.98 | 95.88 | 95.54 | **95.27** |
| CamDist | **44.48** | 46.16 | 49.95 | 52.32 | 54.95 | 59.73 | 57.05 |

As shown in Tables 3 and 4, this movement constraint term sacrifices a little accuracy (mostly within 2) while yielding much better continuity, with consistent decrease in CamDist.

**Supervision signal in reward design:** In Equation (1) we associate the quality term $Q^t$ with minimum joint visibility $\min_k \mathcal{J}_k^t$, which is a self-supervised signal as it does not leverage dataset annotation during training. We can replace this term with a supervised signal

$$
Q^t = 1 - \frac{\varepsilon_k^t}{E}
\tag{4}
$$

where $\varepsilon_k$ is the pose reconstruction error of the $k$-th person, which needs ground truth pose label to calculate. The scaling factor $E$ is set to 200. The supervised signal helps agents to achieve higher accuracy in pose estimation when the number of views is more than 5, without movement constraints in Table 3. In other cases, its performance is close to that of self-supervised signal designed in Equation (1). When movement constraints come into play, supervision signal boosts CamDist as shown in Table 4.

*Prophet* When agents need to work on wilder and more dynamic environments, where movements between frames can be remarkable, the performance can thus be hampered. We change the FPS of our scenes to simulate this scenario, and compare *MAO-Pose* with or without *Prophet* module. The results are provided in Table 5.

**Table 3.** Comparison of mean per joint position error (mm/joint) on the Panoptic datasets. Standard deviation is shown in the bracket. Columns 2–6 indicate the number of agents (or views) we use. Our model outperforms all baselines (except the **Greedy** method). *Supervised* means using supervised signal as the view quality term in the reward. *Cons* means adding movements constraints to penalty term. *noComm* means disabling communication protocol, *Consensus*, between agents.

| Policy | 4 | 5 | 6 | 7 | 8 |
|---|---|---|---|---|---|
| *MAO-Pose* | 113.60 (1.73) | **94.21** (0.86) | 87.53 (0.84) | 84.02 (0.92) | 80.81 (0.34) |
| *MAO-Pose*-Supervised | 116.35 (1.05) | 95.83 (0.60) | **87.44** (0.41) | **83.79** (0.32) | **80.67** (0.12) |
| *MAO-Pose*-Cons | 125.45 (16.87) | 96.09 (0.56) | 88.85 (0.32) | 84.17 (0.43) | 81.87 (0.35) |
| *MAO-Pose*-Cons-Supervised | **110.88** (1.28) | 96.65 (1.17) | 88.19 (0.30) | 84.79 (0.23) | 82.14 (0.27) |
| *MAO-Pose*-noComm | 139.18 (2.80) | 118.02 (2.29) | 102.33 (0.33) | 94.07 (0.49) | 87.47 (0.68) |
| ACTOR [7] | 122.09 (1.39) | 100.55 (1.27) | 89.79 (0.30) | 85.04 (0.29) | 82.24 (0.24) |
| Random | 167.77 (2.12) | 138.05 (1.26) | 117.54 (1.09) | 103.07 (0.59) | 94.58 (0.46) |
| AngleSpread | 171.21 (2.16) | 112.45 (0.25) | 105.47 (1.19) | 93.97 (0.40) | 87.8 (0.33) |
| Greedy | 106.43 | 89.65 | 78.66 | 70.60 | 66.01 |

**Table 4.** Comparison of camera distance (cm) on the Panoptic datasets. Format follows Table 3.

| Policy | 4 | 5 | 6 | 7 | 8 |
|---|---|---|---|---|---|
| *MAO-Pose* | 111.64 (0.57) | 55.92 (0.40) | 64.28 (0.82) | 104.34 (0.49) | 78.56 (0.64) |
| *MAO-Pose*-Supervised | 71.4 (1.05) | 81.99 (0.85) | 79.6 (0.75) | 79.46 (0.55) | 87.36 (0.48) |
| *MAO-Pose*-Cons | 39.20 (0.55) | 38.36 (0.54) | 41.83 (0.24) | 45.05 (0.18) | **42.39** (0.27) |
| *MAO-Pose*-Cons-Supervised | **29.88** (0.48) | **32.91** (0.78) | **36.87** (0.20) | **37.98** (0.56) | 44.40 (0.26) |
| *MAO-Pose*-noComm | 101.55 (0.85) | 84.73 (0.64) | 93.28 (0.32) | 135.07 (0.18) | 96.56 (0.94) |
| ACTOR [7] | 181.02 (1.72) | 134.96 (1.23) | 122.33 (0.34) | 123.06 (0.70) | 126.94 (0.27) |
| Random | 245.29 (0.85) | 220.27 (0.53) | 201.77 (0.84) | 185.84 (0.72) | 172.34 (0.27) |
| AngleSpread | 103.09 (0.34) | 101.94 (0.56) | 102.42 (0.28) | 97.01 (0.60) | 99.61 (0.35) |
| Greedy | 205.52 | 175.83 | 152.62 | 135.41 | 120.94 |

**Table 5.** Ablation study of *MAO-Pose* with or without *Prophet* on the Panoptic test datasets. Columns 3–5 indicate the FPS of the environment. The smaller the FPS is, the more dynamic the environment can be. *Prophet* enables our system to outperform in the dynamic environment while maintaining the smoothness of camera movements.

| Policy | Metric | 10 FPS | 2 FPS | 1 FPS |
|---|---|---|---|---|
| *MAO-Pose* w/o *Prophet* | **MPJPE** | 100.08 | 103.42 | 110.85 |
| | **CamDist** | 59.43 | 61.80 | 64.58 |
| *MAO-Pose* with *Prophet* | **MPJPE** | **97.37** | **101.40** | **107.05** |
| | **CamDist** | **59.27** | **61.12** | **64.53** |

*4.4. Inference Speed*

We conducted the performance test on a server with one Intel E5-2680 V4 and a single GTX 1080Ti GPU. The results are shown in Table 6. Note, we evaluate all the agents jointly, of which the frame rate will be lower than that of agents running independently on different machines in real deployments. In all the settings, the average FPS can exceed our maximum control frequency (10 FPS), denoting that the system can run in *real-time*. The inference speed of each module in MAO-Pose is in Table 7. Note that, 2D pose backbone occupies the majority of time during inference, which indicates that our method is complementary to any 2D pose backbones. Furthermore, one can achieve desired the FPS by choosing a proper 2D pose estimator.

**Table 6.** Performance on all test datasets.

| # Views | 4 | 5 | 6 | 7 | 8 |
|---|---|---|---|---|---|
| Average FPS | 22.38 | 16.92 | 15.25 | 13.11 | 11.35 |

**Table 7.** Inference speed of each module in MAO-Pose.

| Module | Speed (ms) | GFlops |
|---|---|---|
| 2D Pose Network | 55.18 | 7.76 |
| Policy Network (single agent) | 0.78 | 0.017 |
| Communication | 0.79 | - |
| Person Matching Triangulation | 43.99 | - |

*4.5. Proof of Concept*

Our *MAO-Pose* can be easily modified for a multi-drone capture hardware system. A typical application scenario is that people are walking outdoors, and we need to track those moving targets and reconstruct the 3D pose as well. Due to the lack of datasets for 3D pose estimation in the wild, we built up a simulation environment (Figure 8) based on the virtual dataset, *JTA Dataset* [20], which contains massive body poses with full 3D annotations. The *JTA* consists of many full-HD videos that are each 30 s long, recorded at 30 FPS and 1080P resolution with a locked camera. We randomly chose 5 urban scenes, 4 scenes for training and 1 scene for evaluation. Like the training process of *Panoptic*, the scenes are split into parts with no overlap between each other.

In this simulation, each camera is mounted on a flying drone with 6DoF and minimizes the limitation of the dome localization in the Panoptic dataset. The view optimization is proceeded in a virtual dome containing discrete viewpoints similar to Panoptic. We insert a tracking action to move the cameras after the view optimization to keep the target people in the center of the virtual dome.

It is worth noting that there are a limited number of views in the dataset (with only 6 views for each scene). As a consequence, it is hard to extract feature $\mathbf{B}^t$ as the input of policy network of *MAO-Pose* given a view of the agent. Therefore, we manage to generate a *quasi feature* based on 3D annotations to solve the dilemma. Note that, in the simulation environment, our system is still in an annotation-free manner while we only use 3D annotations to simulate features of arbitrary views owing to the limits of the dataset.

Given the pose of the *i*-th drone $P_i = \begin{bmatrix} R_i & T_i \\ 0 & 1 \end{bmatrix}$, where $R_i \in \mathbb{R}^{3 \times 3}$ indicates the rotation and $T_i \in \mathbb{R}^{3 \times 1}$ indicates the translation to the world coordinate. Given camera intrinsic $K \in \mathbb{R}^{3 \times 3}$ and 3D poses $Y \in \mathbb{R}^{3 \times m}$, we can get access to the 2D poses $X_i \in \mathbb{R}^{2 \times m}$ in the view of the *i*-th drone by $Z_i \begin{bmatrix} X_i \\ \mathbf{1} \end{bmatrix} = [K \quad \mathbf{0}] P_i \begin{bmatrix} Y \\ \mathbf{1} \end{bmatrix}$, where $Z_i$ is a scale factor and $m$ is the number of people in current view.

Then, we used the depth of joints as the grayscale value of pixels given the 2D position of joints and skeletons, which aims to encode the spatial information of human bodies. Meanwhile, we computed the occlusion in the current view according to a simple human model, adding noise to the 2D position of occluded joints to encode the occlusions in the quasi feature. In this simulation, the view optimization has proceeded in a virtual dome containing discrete viewpoints similar to the Panoptic dataset. Moreover, we insert a tracking action to move the cameras after the view optimization to keep the target people in the center of the virtual dome, which is applicable in an outdoor scenario. As shown in Figure 8, the virtual dome is drawn in gray lines and each vertex represents the possible camera position. The selected views are displayed in yellow, blue, purple and light green rays.

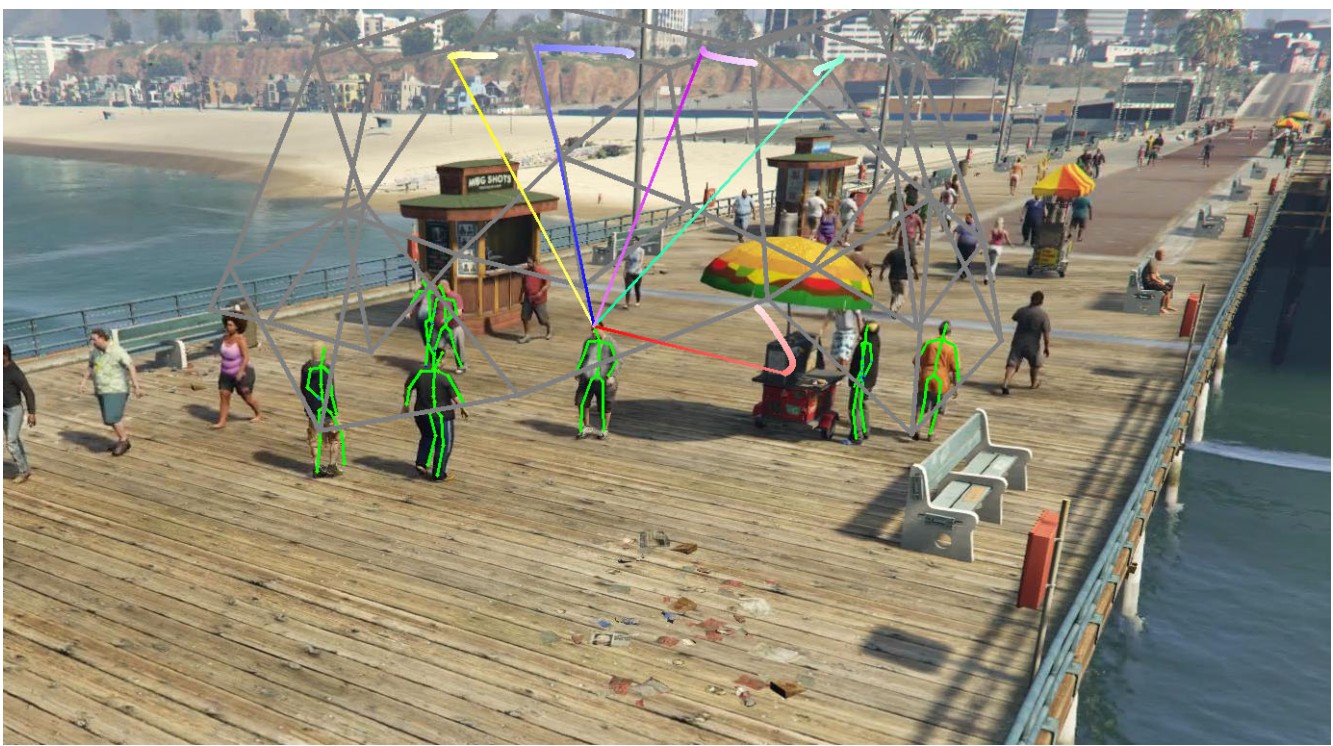

**Figure 8.** *MAO-Pose* operates on simulation environment based on the *JTA* dataset. The virtual dome is drawn in gray, and the cameras selected by agents are all pointing to the target man. Other people within the dome are reconstructed as well.

## 5. Discussion

We proposed an active 3D human poses reconstruction system *MAO-Pose*. In this architecture, each camera is modeled as an agent with individual observation and exchanges the relative position information with an inter-agent communication mechanism, *Consensus*. By maximizing the visibility of dynamic scenario with occlusions, our method could achieve the goal of multi-person pose reconstruction. Towards practical application when cameras mounted on drones, the physical motion constraint is considered in the reward design. We have provided a visualization of the difference between our work and *ACTOR* in Figure 9. Though our work might look ostensibly similar to *ACTOR* because of the Panoptic dataset used in both, our work varies vastly with ACTOR in basic settings and assumptions. MAO-Pose triangulates poses in a single timestep through multiple agents, and thus can capture dynamic scenes in real-time. Contrarily, equipped with just a single agent, *ACTOR* requires multiple timesteps to triangulate poses for a single frame, which can only be used in static scenes (that is why the drone demo reported in [7] requires a person standing still, even keeping static gestures) or post-processing a dynamic scene pre-captured by dense multi-view cameras.

The experiment on the Panoptic dataset shows that *MAO-Pose* achieves more accurate 3D pose reconstruction compared with the baseline method *ACTOR* and is close to Greedy, which is the optimal solution, theoretically. Our method could achieve the lowest moving cost with movement constraints in the penalty term demonstrated by the experiment in frame sequence datasets. Through the ablation studies in different strategies, we also prove that the police with inter-agent communication module in supervised training method could achieve a better performance compared with other police settings.

In the future, *MAO-Pose* should be modified to asynchronous capturing and deployed to real cameras mounted on flying robots, like UAVs, for remote sensing for dynamic environment. We acknowledge that working with real drones indeed involves complex hardware issues, yet our system cannot output parameters intended to directly control real hardware (and, at its base, it is not designed to achieve this objective). Instead, we focused on providing valuable waypoints that were more physically viable through our

policy network, which can later be easily employed by lower-level path planners for real drones. The Panoptic dataset is comprised of discretized camera locations and the camera moving speed is ill-defined on it. When deploying to real-world systems, we might need to incorporate the velocity and/or acceleration into movement constraints. We will also improve the robustness and security of our system for practical implementation situations, such as adding a "Top-View" camera to capture and analyze the whole scenario to avoid the unforeseeable obstacles appearing on the camera way.

This system can be also extended to 3D human pose tracking and active motion capture. The entertainment and sports industry may benefit from our work since the active capture system in this paper helps to better understand human 3D information, which can be applied to filming and sports broadcasting. The photographers may be put at a disadvantage from this research since the camera can learn to select better capture views and thus might replace the job of some photographers. Moreover, the system failure may result in the crash of the cameras and further economic loss. This method does not leverage biases in the data.

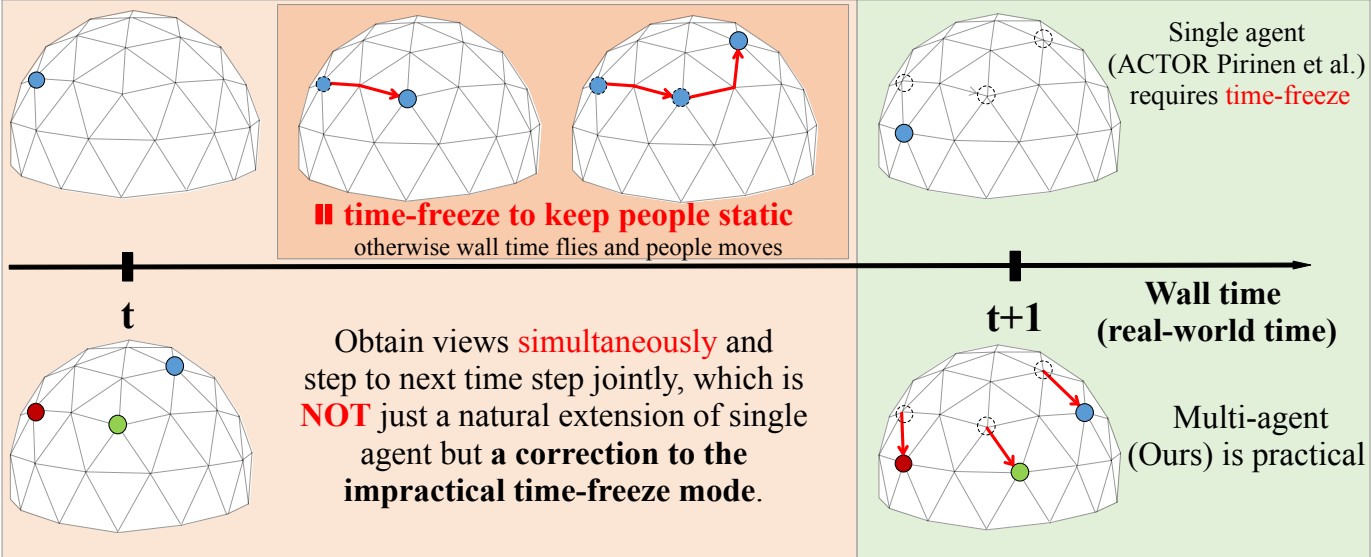

**Figure 9.** Illustration of the difference between ACTOR and our system, *MAO-Pose*.

**Author Contributions:** Conceptualization, Z.F., X.L. and Y.L.; methodology, Z.F. and X.L.; software, Z.F.; validation, Z.F. and X.L.; formal analysis, Z.F., X.L. and Y.L.; investigation, Z.F., X.L. and Y.L.; resources, Y.L.; writing—original draft preparation, Z.F.; writing—review and editing, Z.F., Y.L. and X.L.; supervision, Y.L.; project administration, Y.L.; funding acquisition, Y.L. All authors have read and agreed to the published version of the manuscript.

**Funding:** This work was funded in part by the National Natural Science Foundation of China (U1913602), in part of the National Key R&D Program of China (2018AAA0102804).

**Acknowledgments:** We gratefully thank Xiang Zhang and Zhaoxi Chen for the suggestion of the system method and manuscript editing. We are also grateful to Yuwang Wang for constructive comments during the review process.

**Conflicts of Interest:** The authors declare no conflict of interest.

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
