# Peer review of "Multi-Agent Deep Reinforcement Learning for Online 3D Human Poses Estimation"

_remotesensing, doi:10.3390/rs13193995_

Round 1

Reviewer 1 Report

The authors of the text presented an interesting approach to the Human Pose Estimation topic. The proposed algorithm for pose estimation is clearly described and its results are compared to the other approaches. 
However, in the text I have identified a few problems, that in my opinion must be improved before publication:
- line 22 – please do not start a sentence with a literature reference,
- lines 121-146 and lines 147-172 those two parts of the text are exactly the same - please correct this issue, 
- line 108 – occasions or occlusions? ,
- line 267 – this sentence ends unexpectedly,
- Figure 4 description - unexpectedly the wording 'Oracle' appears, shouldn't it be 'Greedy'?
- line 299 – table 2 or 3?
- line 314 - table 2 or 3?
- Figure 6 – the graphic presentation of the dome is difficult to separate from the background of the drawing. Perhaps the authors will be able to improve the presentation somewhat.

As authors mention in lines 76-77 the statically mounted capture system is inapplicable in outdoor scenarios, but in fact, the arrangement of cameras, e.g. placed on drones, in the shape of a dome, can be also problematic in a real environment. I can imagine the situation that the matrix of UAV is formulated as a planar constellation of devices that follows the moving ‘scene’, whether the algorithm developed by the authors allows for a change to a flatter shape? 

Dear authors please also comment on the possibility of implementation the situation that the camera movement may be limited by obstacles appearing on the camera way. 

Author Response

Dear reviewer:

We provide a point-by-point response to the your comments. Please see the attachment.

Best Regards

Reviewer 2 Report

The paper Multi-Agent Deep Reinforcement Learning for Online 3D
Human Poses Estimation, describe a study for online view selection for a fixed number of cameras to estimate multi-person 3D poses actively, using  distributed multi-agent based deep reinforcement learning framework. Each camera used is model as an agent, to optimize the action of all the cameras 

Some suggestions:

The abstract is well presented

The abreviation CNN must be detailed

Line 22 - the sentences begin with a reference [7, 8] is not very well

Figure must be call in the text with Fig....

What means §3.2

The " This is because with more cameras in the system, the collision probability is higher." must be reformulated without probably

Discussion must be improved with more details about the results

The paper must respect the template

Author Response

(The authors gave the same response as above.)

Round 2

Reviewer 1 Report

Thank you for addressing my comments and correcting the indicated problems in the text.

The authors have corrected the problems identified in Report no.1 and have provided convincing explanations and argumentation in response to my review.

I accept the report presented by the authors and approve the article in present form.